# *Candida auris* in Intensive Care Setting: The First Case Reported in Portugal

**DOI:** 10.3390/jof9080837

**Published:** 2023-08-09

**Authors:** João Henriques, Verónica Mixão, Joana Cabrita, Tiago Isidoro Duarte, Tânia Sequeira, Sofia Cardoso, Nuno Germano, Liliana Dias, Luís Bento, Sílvia Duarte, Cristina Veríssimo, João Paulo Gomes, Raquel Sabino

**Affiliations:** 1Intensive Care Medicine Department, Centro Hospitalar Universitário Lisboa Central, 1150-199 Lisbon, Portugal; joao.henrique@chlc.min-saude.pt (J.H.); joana.cabrita@chlc.min-saude.pt (J.C.); tiago.duarte2@chlc.min-saude.pt (T.I.D.); tania.sequeira@chlc.min-saude.pt (T.S.); sofia.cardoso@chlc.min-saude.pt (S.C.); nuno.germano@chlc.min-saude.pt (N.G.); luis.bento@chlc.min-saude.pt (L.B.); 2Genomics and Bioinformatics Unit, Department of Infectious Diseases, National Institute of Health Doutor Ricardo Jorge (INSA), Av. Padre Cruz, 1649-016 Lisbon, Portugal; veronica.mixao@insa.min-saude.pt; 3Infection Prevention and Control and Antimicrobial Stewardship, Centro Hospitalar Universitário Lisboa Central, 1150-199 Lisbon, Portugal; liliana.dias@chlc.min-saude.pt; 4Innovation and Technology Unit, Department of Human Genetics, National Institute of Health Doutor Ricardo Jorge (INSA), Av. Padre Cruz, 1649-016 Lisbon, Portugal; silvia.duarte@insa.min-saude.pt; 5Reference Unit for Parasitic and Fungal Infections, Department of Infectious Diseases, National Institute of Health Doutor Ricardo Jorge (INSA), Av. Padre Cruz, 1649-016 Lisbon, Portugal; cristina.verissimo@insa.min-saude.pt; 6Faculty of Veterinary Medicine, University of Lisbon, Av. Universidade Técnica, 1300-477 Lisbon, Portugal; 7Instituto de Saúde Ambiental, Faculdade de Medicina, Universidade de Lisboa, 1649-028 Lisbon, Portugal; 8Laboratório Associado TERRA–Laboratório para o Uso Sustentável da Terra e dos Serviços dos Ecossistemas, Instituto Superior de Agronomia, Tapada da Ajuda, 1349-017 Lisbon, Portugal

**Keywords:** *Candida auris*, antifungal resistance, whole-genome sequencing

## Abstract

*Candida auris* is an opportunistic human pathogen that has rapidly spread to multiple countries and continents and has been associated with a high number of nosocomial outbreaks. Herein, we report the first case of *C. auris* in Portugal, which was associated with a patient transferred from Angola to an ICU in Portugal for liver transplantation after a SARS-CoV-2 infection. *C. auris* was isolated during the course of bronchoalveolar lavage, and it was subjected to antifungal susceptibility testing and whole-genome sequence analysis. This isolate presents low susceptibility to azoles and belongs to the genetic clade III with a phylogenetic placement close to African isolates. Although clade III has already been reported in Europe, taking into account the patient’s clinical history, we cannot discard the possibility that the patient’s colonization/infection occurred in Angola, prior to admission in the Portuguese hospital. Considering that *C. auris* is a fungal pathogen referenced by WHO as a critical priority, this case reinforces the need for continuous surveillance in a hospital setting.

## 1. Introduction

*Candida* spp. infections have progressively increased over the past decades and become a major cause of morbidity and mortality in critical care patients. This phenomenon is mainly associated with the increasing rate of invasive procedures, the extensive use of broad-spectrum antimicrobials, the higher rate of immunocompromised patients admitted to Intensive Care Units (ICUs), and the length of ICU stay [1]. Although *C. albicans* accounts for the majority of *Candida* infections, several non-*albicans Candida* species have been associated with an increasing incidence of invasive infection with high rates of therapeutic failure [2].

In 2022, the WHO launched the first-ever list of priority pathogens (WHO FPPL) as the first global effort to systematically prioritize fungal pathogens, considering their unmet research and development (R&D) needs and perceived public health importance. Of the 19 fungi that represent the greatest threat to public health, *Aspergillus fumigatus*, *Candida albicans*, *Cryptococcus neoformans*, and *Candida auris* were considered as of critical priority [3]. *C. auris* was first reported in Japan in 2009, and since then, it has emerged worldwide, being associated with nosocomial outbreaks in intensive care settings. In Europe, according to the European Centre for Disease Prevention and Control (ECDC), *C. auris* has been identified in more than a dozen countries. The last outbreak was reported in 2019–2021 in northern Italy with at least 277 cases, some of them in patients with severe COVID-19 infections [1,4]. Between 2013 and 2021, 1812 cases of *C. auris* infections or carriage were reported by 15 EU/EEA countries, and the number of reported cases almost doubled from 2020 (335 cases) to 2021 (655 cases) [5]. In the USA, the number of *C. auris* cases increased by 44% in 2019 and by 95% in 2021 and the number of cases resistant to echinocandins in 2021 was about three times higher than in each of the two previous years [6]. 

Public health concerns regarding this infection occur because it combines critical characteristics, such as the potential to spread through horizontal transmission, the ability to cause life-threatening infections, the multi-resistance profile, and the limitations for optimal treatment [7,8]. The crude in-hospital mortality rate for *C. auris* candidemia is estimated to range from 30 to 72%, even in patients receiving antifungal treatment [1]. Higher rates are associated with patients admitted to ICU and with candidemia, although these may be mainly attributable to the severity of the underlying diseases [4,7].

Given the clinical importance and global spread of this species, we report and characterize the first case of *C. auris* identified in Portugal, including a brief phylogenetic study of this isolate.

## 2. Case Presentation

A 59-year-old male from Angola was admitted to a hospital in Luanda, and 10 days prior to admission, he had a positive SARS-CoV-2 antigen test with mild symptoms. He presented rapid progressive cough, shortness of breath, and fatigue, and was then transferred to the hospital Intensive Care Unit (ICU) with respiratory failure and cardiogenic shock. After a short period of invasive ventilatory and vasopressor support, he developed multiorgan failure with acute kidney injury and presumed acute liver failure (ALF), as he had hyperbilirubinemia, cytolysis, and coagulopathy. 

Considering the possible eligibility for liver transplantation, the patient was transferred to Lisbon, Portugal, and admitted directly to the ICU in a hospital, which is a reference center in transplantation. At admission, he was under sedation, with invasive mechanical ventilation and vasopressor support with noradrenaline. Continuous renal replacement therapy (CRRT) was prescribed, coagulopathy was corrected with blood products transfusions, and he initiated Ceftriaxone as empirical treatment for presumed infection with an elevation of inflammatory markers. He was immediately submitted for trans-jugular liver biopsy in order to investigate the etiology of ALF. Hemodynamic evaluation with echocardiogram showed reduced ejection fraction (20%), with an estimated cardiac index of 2.8 L/m^2^/min. The clinical presentation did not fulfill the criteria for liver transplantation. In fact, the investigation on infectious and autoimmune causes for acute hepatitis was negative, and the biopsy revealed necrotic liver tissue, leading the physicians to assume the diagnosis of ischemic hepatitis due to cardiogenic shock. The investigation went on, and etiologies for myocardiopathy as a cause of cardiogenic shock were searched. The coronary angiogram was normal, and all potentially infectious agents were excluded (the diagnostic tests included various real-time PCR for SARS-CoV-2). There was a consensus on the diagnosis of viral myocardiopathy despite the fact the proper virus could not be identified.

During his stay in the ICU, the patient evolved in sudden primary severe acute respiratory distress syndrome (ARDS) due to ventilator-associated pneumonia with a need for extracorporeal membrane oxygenation (ECMO) support. Multidrug-resistant (MDR) *Acinetobacter baumanii* was isolated in bronchoalveolar lavage (BAL) and blood cultures, and he was treated with colistin, according to the antimicrobial susceptibility test (AST). The patient had support with veno-venous ECMO for four days. After decannulation, there was a clinical reaggravation with septic shock. Before these clinical findings, after 4 days of treatment with colistin for pneumoniae and bacteriemia with *A. baumannii*, another BAL was performed, in which, in addition to *A. baumannii*, *C. auris* was also isolated. 

Despite the continued treatment with colistin for 16 days and also with caspofungin for 10 days, there was progressive clinical deterioration with ARDS, refractory shock, and multiorgan failure. Even with all organ support and treatments, the patient died one month after initial admission in the ICU in Luanda with no other plausible reason to justify the multiorgan failure except the lack of response of both etiological agents to the treatment.

## 3. Materials and Methods

The isolate CaurisPT_1_2022 was identified as *C. auris* by MALDI-TOF_MS at the hospital laboratory and was then sent to the Mycology Reference Laboratory at the Portuguese National Institute of Health (INSA) for further tests, including molecular identification for species confirmation, determination of the antifungal susceptibility profile, and genomic analysis.

Colonies were isolated in Sabouraud dextrose agar supplemented with cloranphenicol (0.05%). Total genomic DNA was extracted from the purified colonies. For species confirmation, the D1–D2 region of 28S ribosomal DNA (rDNA) was amplified using the primer set NL1 and NL4. Sanger sequencing of both strands was performed and nucleotide sequences were edited using the program Chromas Lite v 2.01 and aligned with the program CLUSTALX v 2.1 [9]. A BLASTn was performed in GenBank (Bethesda, MD, USA) and CBS-KNAW Fungal Biodiversity Centre databases (Utrecht, the Netherlands) for highly similar sequences (megablast) and to all CBS databases, respectively. 

Antifungal susceptibility was performed by microdilution using the MICRONAUT-AM AFST system (MERLIN Diagnostika GmbH, Bornheim, Germany), according to manufacturer’s instructions, and minimal inhibitory concentrations (MICs) were determined according to EUCAST procedures. The antifungals included in this test were anidulafungin (0.002 mg/L, 0.015–8 mg/L), micafungin (0.002 mg/L, 0.015–8 mg/L), caspofungin (0.002 mg/L, 0.015–8 mg/L), fluconazole (0.002 mg/L, 0.25–128 mg/L), posaconazole (0.0078–8 mg/L), voriconazole (0.0078–8 mg/L), itraconazole (0.031–4 mg/L), amphotericin B (0.031–16 mg/L), and 5-flucytosine (0.0625–32 mg/L).

For whole-genome analysis, total DNA was subjected to Nextera XT library preparation and paired-end sequenced on an Illumina NextSeq 2000 (2× 150 bp), according to manufacturer’s instructions. Sequencing reads were inspected with FastQC v0.11.9 [10] and filtered with Trimmomatic v0.39 [11]. A K-mer Analysis Toolkit (KAT) was used to count the *k*-mer frequency and GC content and to estimate the expected genome size [12], using default parameters. SPAdes v3.13.0 was used to perform the genome assembly [13]. The quality of the assembly was inspected with Quast v5.2.0 [14]. As contamination with *Cellulosimicrobium cellulans* was detected, KAT was used to obtain the *k*-mer hashes that corresponded to the possible contaminant and to remove their contigs in the assembly. A final step of removal of redundant contigs and scaffolding was performed with Redundans [15].

In order to place the *C. auris* isolate among the previously defined genetic clades of this species, a comparative genomics analysis including other 24 *C. auris* samples was performed. These samples were carefully selected from a recent population genomics study of this pathogen to ensure the representativeness of clades I–IV of multiple continents (including all available European isolates) and of different antifungal-resistance profiles (Appendix A) [16]. Whole-genome sequencing (WGS) data were retrieved from the Sequence Read Archive (SRA). Read mapping and variant calling were performed with the HaploTypo pipeline v1.0.1, using default parameters [17]. Briefly, BWA-MEM 0.7.17-r1188 was used to align the sequencing reads in the reference genome of *C. auris* (BP8441; accession: GCA_002759435.2) [18]. GATK 4.2.6.1 was used to perform variant calling considering a minimum depth of coverage of 20 reads and a haploid genome [19]. The genomic sequence of each sample was then obtained with GATK FastaAlternateReferenceMaker tool. Positions with INDELs or covered by less than 20 reads in at least one sample were excluded. SNP-sites was used to obtain the variant sites [20]. Maximum-likelihood tree reconstruction of the alignment comprising 193,870 variant sites was performed with RAxML v8 [21]. Following the same procedure, after the association of the Portuguese *C. auris* isolate to clade III, we retrieved all isolates of this clade used in the above-mentioned population genomics study from SRA [16] and reconstructed an additional maximum-likelihood tree considering only these isolates (and the Portuguese sample) and their 3278 variant sites.

## 4. Results

### 4.1. Molecular Identification and Antifungal Susceptibility

The obtained sequences from the D1–D2 region were compared with sequences deposited in the GenBank and CBS-KNAW Fungal Biodiversity Centre databases. In GenBank, the similarity and coverage were 99.62% and 100%, respectively, with the sequence accession number CP043535; in the CBS-KNAW Fungal Biodiversity Centre database, the similarity and coverage were 99.80% and 97.70%, respectively, with the sequence accession number CBS 15615_ex94611_LSU.

The susceptibility profile of the isolate, CaurisPT_1_2022, is reported in Table 1.

### 4.2. Whole-Genome Analysis

The genomic characterization of CaurisPT_1_2022 was performed with a comparative genomics approach including other publicly available samples (Appendix A and Methods for more details). Our results showed that this isolate belongs to clade III, which has previously been associated with Africa, but also enrolls strains from different continents, including Europe (Figure 1A) [16,23]. Indeed, the three strains sequenced so far in Spain also belong to the same clade. Nevertheless, the Portuguese isolate seems to be more closely related to isolates from Africa than to the European ones (Figure 1B). Furthermore, we assessed the presence of non-synonymous mutations of the potential association with antifungal resistance and detected the presence of C125A and F126L mutations in *ERG11*, both of them potentially associated with azole resistance (Appendix A) [24]. These results are in agreement with the phenotypic susceptibility tests. 

## 5. Discussion

Due to climatic changes and temperature increases, new pathogenic fungi that cause invasive fungal diseases are emerging, and the yeast *C. auris*, although only described in 2009, is now boasting a near-global distribution [25]. *Candida auris* is an emerging fungus that presents a serious global health threat; it is often multidrug-resistant and frequently causes impacting outbreaks in healthcare settings. For this reason, it is important to quickly identify *C. auris* in a hospitalized patient so that healthcare facilities can take special precautions to stop its spread [22]. 

The identification of the *C. auris* reported in the present study is consonant with other reported cases in the literature [26]. In fact, it happened in a critically ill patient, who presented multiple organ failure and had a prolonged length of stay in ICU. This patient underwent a large range of invasive techniques and multiorgan support, such as invasive mechanical ventilation, CRRT, ECMO, and transfusions, among others. Each of these procedures was likely to have contributed to a worse impact on the susceptibility to infection and outcome in terms of mortality. In the present case, the detection of *C. auris* in BAL happened on day 4 of treatment with colistin for *A. baumannii* according to AST, and it coincided with a clinical reaggravation after ECMO decannulation, with septic shock. Despite bacteriemia and persistent pneumoniae due to *A. baumanni*, the *C. auris* isolate could justify the clinical worsening. Although the plausible hypothesis of colonization with *C. auris* in a critically ill patient transferred from Angola, with risk factors for fungal infection, this isolate was considered to be an etiological agent of infection by the clinicians. The therapeutical decision of giving caspofungin, though, did not change the final outcome. The literature supports the high mortality rate associated with *C. auris* infection [1], but in this case, it remains unclear the role of *C. auris* in clinical aggravation with multiorgan failure and death. In fact, there was a co-infection with *C. auris* and *A. baumannii*, with both microorganisms associated with high mortality [1,8,27], which cannot be neglected, but what is unclear is its influence on the patient’s outcome, as there were 16 days of proper treatment with colistin according to AST and 10 days with caspofungin. Despite the reported association between *C. auris* and other bloodstream infections and the use of broad-spectrum antibiotics [1], there is a lack of literature with no other cases of co-infection with *A. baumannii* reported. On the other hand, and despite the negative real-time PCR, the previously reported COVID-19 infection is noteworthy. Indeed, although there was no identification of the proper virus, there was a consensus on the diagnosis of a viral myocardiopathy likely caused by SARS-CoV-2. In addition, there are other cases of *C. auris* described in COVID-19 patients [28], and there is no information regarding their susceptibility to *C. auris* infections.

Since there was previous isolation of an MDR *A. baumannii*, measures to stop the spreading of this agent were already taken into account: the patient stayed in an isolation room, whose staff were using personal protective equipment; the patient was already under surveillance by the local Infection Prevention and Control and Antimicrobial Stewardship when *C. auris* was isolated. Then, the safety measures were reinforced, with regular sanitizing of the whole ICU, which successfully avoided its transmission, with no other isolates of *C. auris* reported.

No antifungal susceptibility breakpoints for *C. auris* are currently standardized for the European Committee on Antimicrobial Susceptibility Testing (EUCAST) or the Clinical and Laboratory Standards Institute (CLSI). However, the Centers for Disease Control and Prevention (CDC) defined tentative resistance breakpoints, as indicated in Table 1. According to these values, our isolate is resistant to only one antifungal class (azoles), mirroring other reports [29]. Genomic analysis revealed the presence of genetic markers known to be associated with this phenotype. It also placed this isolate among *C. auris* clade III, which has already been reported in Europe [30]. Curiously, the closest sequences to this isolate are not the available Spanish sequences, but rather sequences from Africa. Considering that this patient was initially admitted to an ICU in Angola before being transferred to the ICU in Portugal, one cannot discard the hypothesis that the infection/colonization took place in the Angola hospital, which is likely characterized by the presence of populations of microbial pathogens considerably different from the ones we are used to dealing with in our ICU. Nevertheless, population genomic studies corroborate the global spread of *C. auris*, considering the phylogenetic clustering of strains with a diverse geographical origin [16]. The development of rapid genomic analysis has been key to understanding the international and local-scale transmission of *C. auris*, including the emergence of multidrug-resistant variants [31].

## 6. Conclusions

To date, this is the first and single case of *C. auris* identified in Portugal. However, due to the multiple comorbidities and the identification of other infecting agents, the role of *C. auris* in the patient’s disease progression and outcome cannot be disclosed. Considering the genomic data and the hospital admission history, the African origin for this *C. auris* isolate is plausible. This case reinforces the need for continuous surveillance in the hospital setting and for a better understanding of the risk factors, the mechanisms of virulence, and preventive and treatment measures. 

## Figures and Tables

**Figure 1 jof-09-00837-f001:**
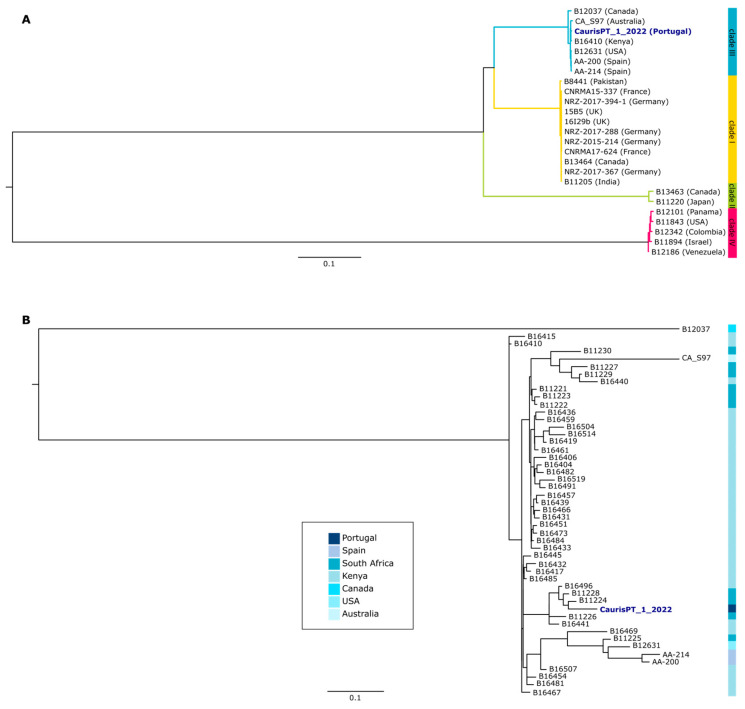
Phylogenetic characterization of the Portuguese isolate of *C. auris*. (**A**) Maximum-likelihood tree reconstruction of the multi-sequence alignment of 25 *C. auris* isolates. Countries of origin of each isolate are indicated in parentheses. Clade 1 is highlighted in yellow, clade II in green, clade III in blue, and clade IV in pink. (**B**) Maximum-likelihood tree reconstruction of the multi-sequence alignment of 49 *C. auris* isolates of clade III, with the country of origin of each isolate being indicated according to the blue color scale. Both trees were rooted with the midpoint rooting method.

**Table 1 jof-09-00837-t001:** Susceptibility profile of the isolate, CaurisPT_1_2022, and comparison with CDC tentative resistance breakpoints [22].

Antifungal	MICs CaurisPT_1_2022 Isolate (mg/L)	CDC Tentative Resistance Breakpoints (mg/L)
Fluconazole	>128	≥32
Itraconazole	>4	Consider using fluconazole susceptibility as a surrogate for second-generation triazole susceptibility assessment
Voriconazole	2	
Posaconazole	0.25	
Amphotericine	0.25	≥2
Anidulafungun	0.25	≥4
Micafungin	0.0625	≥4
Caspofungin	0.25	≥2
5-fluorocytosine	<0.0625	Not determined

## Data Availability

The obtained sequence of the isolate, CaurisPT_1_2022, was deposited in Genebank with the accession number OQ349558. Whole-genome sequencing data are available at the European Read Archive (ENA) under BioProject PRJEB64175.

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
