# Peer review of "Candida auris in Intensive Care Setting: The First Case Reported in Portugal"

_jof, 2023, doi:10.3390/jof9080837_

Round 1
Reviewer 1 Report
João Henriques et al reported the first Candida auris in an Intensive Care setting in Portugal. This communication is timely and impressive. Given the high in-hospital mortality rate for C.auris systemic infection, more caution should be given to this species. I have some concerns:
1. Considering the spread speed of C.auris in clinical settings, is any action taken to look for any infection in the other patients living in the same ICU unit? And also, any isolation from the near environment? Such as the venous tubes?
2. How about the pathogenicity of this isolate? Hypovirulence or hyper virulence?
Author Response
João Henriques et al reported the first Candida auris in an Intensive Care setting in Portugal. This communication is timely and impressive. Given the high in-hospital mortality rate for C. auris systemic infection, more caution should be given to this species. I have some concerns:
Dear Reviewer 1, thank you for your interest in this communication. Your comments and questions are very pertinent, and we hope to better clarify these issues in the reviewed manuscript.
- Considering the spread speed of auris in clinical settings, is any action taken to look for any infection in the other patients living in the same ICU unit? And also, any isolation from the near environment? Such as the venous tubes?
This patient was already under surveillance by the local Infection Prevention and Control and Antimicrobial Stewardship because of the infection by MDR Acinetobacter baumannii, as it is now explained in lines 235-240. In fact, he was already staying in an isolation room facility in which the healthcare staff were aware of the need of using personal protective equipment, even before the isolation of Candida auris. After C. auris isolation, the hospital Infection Prevention and Control and Antimicrobial Stewardship re-evaluated the isolation measures in course and gave instructions to the ICU staff. These indications included regular cleaning and disinfection of the whole ICU during the patient’s stay and after discharge. As there were not close contact of this patient with other patients, those were not screened. All indications from the hospital control of infection group were rigorously performed, including screening samples from surfaces of the surrounding environment. Until this date, no other isolate of C. auris was detected in the ICU.
- How about the pathogenicity of this isolate? Hypovirulence or hyper virulence?
The WGS analysis of our isolate CaurisPT_1_2022 showed that it encodes orthologs of known virulence factors in C. albicans, including genes associated with biofilm formation, adherence, antifungal drug resistance, stress resistance or production of secreted aspartyl proteases (SAPs), and other enzymes associated with virulence attributes, similarly to the reference genome. However, as transcriptome analysis was not performed to test the expression of those genes associated with these virulence attributes, nor in vitro or in vivo studies were done, we cannot take any conclusion about its virulence phenotype. Therefore, we opted to do not refer this topic in the manuscript, having only mentioned the observations regarding antifungal resistance, as antifungal susceptibility testing was performed and this isolate revealed resistance to azoles, as shown by the detection of C125A and F126L mutations in ERG11 gene.
Reviewer 2 Report
Henriques et al. report the first case of C. auris in a patient who was admitted into a Portuguese hospital after a transfer from Angola to Portugal. Initially, the patient was admitted to ICU awaiting liver transplantation which was later canceled to due the general clinical presentation of the patient. He suffered from ventilator-associated pneumonia and during ECMO support, Acinetobacter baumannii and Candida auris were isolated from bronchoalveolar lavage. Despite treatment with Caspofungin, the overall condition of the patient worsened, leading to multi-organ failure and ultimately death. The authors performed antifungal susceptibility testing and whole genome sequencing with the isolated C. auris strain. The isolate was found to be resistant against fluconazole and could be placed into clade III of C. auris.
The case presentation is clear and easy to understand even for non-clinical experts. The phylogenetic analysis and the antifungal susceptibility testing are technically sound. Within the discussion the authors explain that it is likely that the patient was already infected with C. auris in Angola rather than in Portugal. This supported by the genome sequencing results.
Overall, the manuscript shows again that especially transport of patients between countries can contribute to the worldwide distribution of C. auris. As aforementioned, the presented work is technically sound.
Some points should nevertheless be addressed prior to a publication:
1) It remains unclear whether the patient died of a C. auris infection or due to the infection with A. baumanni or just due to the general health problems. This should be clarified.
2) Is the detection of C. auris in the BAL really an infection? Or is it just colonization? Although I would agree that C. auris in BAL combined with ECMO is a clinical problem, the authors are encouraged to discuss the question of colonization and infection. Especially, at it is also unclear if C. auris really contributed to the patient’s death.
3) Was there treatment for the infection with A. baumanni? And if yes, which one?
4) I think the authors could discuss the interesting co-infection with C. auris and A. baumannii. Are there other cases known and what did it mean for the infected patients?
5) The authors explained that this is the first C. auris case in Portugal. Is there a kind of awareness for C. auris in Portuguese hospitals and are possible cases to be reported to some central laboratory (e.g. national reference center, health authorities). The question is if this the first case because it is something new or is C. auris already presented in Portugal but was not reported so far because no infections occurred or nobody looked for it?
6) Was the nursing staff who took care of the patient screened for C. auris to avoid future transmission to other patients? How was the overall hygiene management in the ICU to avoid transmission, e.g. how were possible contaminated medical device disinfected?
In general, the manuscript is well-written. However, some tiny mistakes and typos should be corrected. For example, it should be Acinetobacter baumannii in line 104.
Author Response
Henriques et al. report the first case of C. auris in a patient who was admitted into a Portuguese hospital after a transfer from Angola to Portugal. Initially, the patient was admitted to ICU awaiting liver transplantation which was later canceled to due the general clinical presentation of the patient. He suffered from ventilator-associated pneumonia and during ECMO support, Acinetobacter baumannii and Candida auris were isolated from bronchoalveolar lavage. Despite treatment with caspofungin, the overall condition of the patient worsened, leading to multi-organ failure and ultimately death. The authors performed antifungal susceptibility testing and whole genome sequencing with the isolated C. auris strain. The isolate was found to be resistant against fluconazole and could be placed into clade III of C. auris.
The case presentation is clear and easy to understand even for non-clinical experts. The phylogenetic analysis and the antifungal susceptibility testing are technically sound. Within the discussion the authors explain that it is likely that the patient was already infected with C. auris in Angola rather than in Portugal. This supported by the genome sequencing results.
Overall, the manuscript shows again that especially transport of patients between countries can contribute to the worldwide distribution of C. auris. As aforementioned, the presented work is technically sound.
Dear Reviewer 2, thank you for your comment and pertinent questions. In fact, it enlightened us to clarify some aspects regarding the clinical case. Therefore, we developed a more detailed discussion on the paper that we hope that brings the answers to the raised topics. Additionally, we present a pint-by-point explanation on each topic mentioned.
Some points should nevertheless be addressed prior to a publication:
- It remains unclear whether the patient died of a auris infection or due to the infection with A. baumanni or just due to the general health problems. This should be clarified.
At the moment of death, the patient had a poor clinical condition, with weakness due to prolonged stay in the ICU and ECMO included, and we attributed the death to a refractory septic shock probably caused by both Acinetobacter baumannii and Candida auris. Despite the treatment with 16 days of Colistin and 10 days of caspofungin, the patient’s clinical condition did not improve and the patient died with no other plausible reason to justify the multiorgan failure. Other biological samples were collected but no other microorganisms were identified. The last paragraph of the Case presentation aims to clarify this topic (lines 109-114).
- Is the detection of auris in the BAL really an infection? Or is it just colonization? Although I would agree that C. auris in BAL combined with ECMO is a clinical problem, the authors are encouraged to discuss the question of colonization and infection. Especially, at it is also unclear if C. auris really contributed to the patient’s death.
Thank you for this question. The clinical evolution supports the thesis of C. auris infection rather than colonization. After identifying A. baumannii in the first BAL and starting Colistin under ECMO, there was clinical improvement which allowed weaning from ECMO support. However, a few days later, there was new worsening with the identification in the second BAL of A. baumannii and C. auris. Due to this reaggravation associated with the new isolate, the medical team did not consider C. auris a colonization but an infection. This discussion is now developed in lines 213-219.
- Was there treatment for the infection with baumanni? And if yes, which one?
All Acinetobacter baumannii isolates, from blood cultures and the two from bronchoalveolar lavage, showed susceptibility only to colistin. There was no evidence suggesting that treatment for the infection from A. baumanni should not be applied. Data about the treatment of A. baumanni can be found in lines 101-103; 213-215; 225-226.
- I think the authors could discuss the interesting co-infection with auris and A. baumannii. Are there other cases known and what did it mean for the infected patients?
Thank you for encouraging us on the discussion of this particular topic. As you can now notice in lines 223-229, discussion on this was added to the manuscript. Co-infection caused by A. baumannii and C. auris cannot be neglected because these microorganisms are both associated with high mortality rates and affect critically ill patients from the ICU. In fact, Ibrahim S. et al report mortality rates up to 70% in ventilator-associated pneumonia by A. baumannii (doi: 10.1007/s11033-021-06690-6), and A. Jeffery-Smith et al (doi: 10.1128/CMR.00029-17) mention mortality rates up to 72% caused by C. auris. By reviewing the literature, we did not find any other reports of co-infection with A. baumannii and C. auris and therefore, the true impact of this co-infection on the patient’s outcome remains unclear.
- The authors explained that this is the first auris case in Portugal. Is there a kind of awareness for C. auris in Portuguese hospitals and are possible cases to be reported to some central laboratory (e.g. national reference center, health authorities). The question is if this the first case because it is something new or is C. auris already presented in Portugal but was not reported so far because no infections occurred or nobody looked for it?
There is awareness about this etiological agent of infection. The Mycology reference laboratory has launched several news, communications and reports emphasizing the emergence of this pathogenic species. In 2020, the Portuguese reference center also launched a control quality assessment program in order to perceive if the Portuguese laboratories were able to identify this species. This program allowed to enhance the awareness on this issue in all participant laboratories. Since then, this control quality assessment scheme was implemented and it is distributed annually to every laboratory that want to participate. Since the isolation of C. auris does not configure a case of mandatory reporting, we cannot rule out that a previous strain may have been isolated and not reported. However, since several years ago, the Mycology reference laboratory has been receiving isolates suspected to be C. auris in order to perform confirmatory tests. All those isolates were sequenced (D1-D2 and ITS regions) and until the case described in this study, none of the sent isolates were confirmed to be C. auris (the majority of them belonged to the C. haemulonii complex).
- Was the nursing staff who took care of the patient screened for auris to avoid future transmission to other patients? How was the overall hygiene management in the ICU to avoid transmission, e.g. how were possible contaminated medical device disinfected?
As it is now explained in lines 235-241, safety isolation measures and the use of personal protective equipment were already in course before C. auris detection, because of the MDR A. baumannii infection. Nevertheless, when C. auris was identified, the local Infection Prevention and Control and Antimicrobial Stewardship gave instructions to the ICU staff regarding infection control measures for prevention of C. auris transmission, based on the Centers for Disease Control and Prevention (CDC) indications. The patient was staying in an isolation room; the care to this patient by the nursing staff was favored to be single restricted; and profound cleaning and disinfection routines were performed regularly to all ICU, during the patient’s stay and also after discharge. There were not any indications to screen the healthcare staff.
Round 2
Reviewer 1 Report
Can be accepted.
Reviewer 2 Report
I thank the authors for addressing my concerns. I am fine with the revised mansuscript.